# Prediction of Safety Risk Levels of Veterinary Drug Residues in Freshwater Products in China Based on Transformer

**DOI:** 10.3390/foods11121690

**Published:** 2022-06-09

**Authors:** Tongqiang Jiang, Tianqi Liu, Wei Dong, Yingjie Liu, Cheng Hao, Qingchuan Zhang

**Affiliations:** 1National Engineering Research Centre for Agri-Product Quality Traceability, Beijing Technology and Business University, Beijing 100048, China; jiangtq@btbu.edu.cn (T.J.); 2031101006@st.btbu.edu.cn (T.L.); liurita0911@163.com (Y.L.); hc2680400194@163.com (C.H.); zqc1982@126.com (Q.Z.); 2School of E-Business and Logistics, Beijing Technology and Business University, Beijing 100048, China

**Keywords:** risk assessment, veterinary drug residues, freshwater products, safety risk-level prediction, transformer

## Abstract

Early warning and focused regulation of veterinary drug residues in freshwater products can protect human health and stabilize social development. To improve the prediction accuracy, this paper constructs a Transformer-based model for predicting the safety risk level of veterinary drug residues in freshwater products in China to conduct a comprehensive assessment and prediction of the three veterinary drug residues with the maximum detection rate in freshwater products, including florfenicol, enrofloxacin and sulfonamides. Using the national sampling data and consumption data of freshwater products from 2019 to 2021, this paper constructs a self-built dataset, combined with the k-means algorithm, to establish the risk-level space. Finally, based on a Transformer neural network model, the safety risk assessment index is predicted on a self-built dataset, with the corresponding risk level for prediction. In this paper, comparison experiments are conducted on the self-built dataset. The experimental results show that the prediction model proposed in this paper achieves a recall rate of 94.14%, which is significantly better than other neural network models. The model proposed in this paper provides a scientific basis for the government to implement focused regulation, and it also provides technical support for the government’s intervention regulation.

## 1. Introduction

With the improvement of people’s living standards, freshwater products are rich in protein, calcium and other nutrients, and they have gradually become an essential food on people’s tables. Moreover, as a large country in the freshwater industry, the export of freshwater products plays an important role in China’s foreign trade, so the quality and safety of freshwater products is not only related to people’s lives and health, but also related to the influence and status of China in international foreign trade. Therefore, the quality and safety supervision of freshwater products is very important [1,2]. However, as the scale of aquaculture continues to expand, the density of aquaculture is also increasing, and the outbreak of various aquatic diseases is becoming more frequent, resulting in the production process being prone to excessive drug use, not in accordance with the provisions of the drug and the use of prohibited drugs [3]. Drug abuse can cause drug residues in the bodies of aquaculture animals, and drug residues on humans and the environment are mainly chronic, long-term and cause cumulative harm. Research reports that studies have shown that some drug residues can produce carcinogenic effects, mutagenic effects, teratogenic effects, developmental toxicity, accumulation in the body, immunosuppression, sensitization and the induction of drug-resistant strains of bacteria [4].

The frequent occurrence of major food safety incidents in the country has had a bad impact on the international arena, not only posing a serious threat to the physical and mental health of consumers, but also causing an incalculable impact on China’s foreign trade in food [5], so it is crucial to monitor the quality and safety risks of domestic aquatic products, which is related to the safety of domestic people’s livelihood, as well as China’s international economic status. Through freshwater product safety risk assessment and early warning analysis, we can provide scientific means for domestic market supervision, on the one hand, and provide safety assurance for healthy freshwater fish farming and consumers’ peace of mind, on the other hand.

Zhang et al. [6] described the rationale and role of risk assessment, summarized the process of veterinary drug residue risk assessment, outlined the qualitative and quantitative risk assessment methods used in the field and proposed the establishment of a new regulatory model for meat safety to improve the existing regulatory system for meat safety. Alan et al. [7] pointed out that JECFA comprehensively addressed both acute and chronic risks through corresponding estimates of acute and chronic exposures and appropriate correction of the Gallo–Torres model for limited bioavailability of bound residues. Wei et al. [8] ranked the risk matrix for veterinary drug residues using the Council on Veterinary Drug Residues and discussed the types of high risk for veterinary drug residues. Demetra et al. [9] studied the dietary exposure assessment of veterinary antibiotics in pork consumed by children and adolescents in Cyprus and found that antibiotic residues in pork were below the allowable daily intake (ADI), and the risk to human health from antibiotic exposure was low based on the estimated daily intake (EDI) of veterinary antibiotics.

The existing literature mainly uses dietary exposure, the food safety index or the target hazard quotients (THQ) method and other individual indicators for risk assessment of veterinary drug residues in edible products, without comprehensive use of existing consumption data, safety intake data, sampling data and other comprehensive assessments and gradings of veterinary drug residues in freshwater products in multiple provinces across the country.

In recent years, due to its ability to analyze historical data of dynamic systems and predict future operating patterns [10], a time series analysis has been commonly used in weather forecasting [11], earthquake precursor forecasting [12], and crop pest and disease hazard forecasting [13], a feature that also meets the requirements of food safety risk prediction. Jiang et al. [14] used hyperspectral discrete wavelet transformation and deep learning to detect and identify veterinary drug residues in beef. Wang et al. [15] predicted the risk hazard of heavy metals in processed grain products using a voting integrated deep learning approach. Jiang et al. [16] used deep learning to grade and predict the safety risk level of carbofuran pesticide residues in Chinese vegetables.

After the statistics of the State Administration for Market Regulation Statistics sampling data, it was found that in freshwater products, 7 out of 16 drugs are detected, among which florfenicol, enrofloxacin and sulfonamides have the highest detection rates and are much higher than the residues of other veterinary drugs. The detection method used in this study is liquid chromatography-tandem mass spectrometry.

In summary, this paper investigates the national sampling data of veterinary drug residues in freshwater products from 2019 to 2021, and it selects the three veterinary drug residues with the maximum detection rate in freshwater products for assessment and prediction, including florfenicol, enrofloxacin and sulfonamides. In this paper, using the national sampling data of veterinary drug residues in freshwater products from 2019 to 2021 and the weekly consumption data of freshwater products in each province, a safety risk assessment model for freshwater products is constructed, and the indicators in the safety risk assessment model are calculated so as to complete the construction of the self-built dataset, combined with the k-means algorithm to classify the weekly freshwater products in each province into risk levels and establish the risk-level space. Finally, based on the Transformer neural network model, the safety risk assessment indicators of freshwater products in each province are predicted based on the self-built dataset, and the freshwater products in that province for that week are classified into the corresponding risk-level space. The model proposed in this paper not only provides a systematic risk measure for the government, but it also provides a scientific reference basis for confirming the priority regulatory order in regulation, and it provides technical support for the government’s intervention in regulation.

## 2. Materials and Methods

### 2.1. Materials

#### 2.1.1. Data Sources

The data of freshwater products in this study were obtained from the sampling data of the National Food Safety Administration from 2019 to 2021, covering 20 provinces, where freshwater products included freshwater fish, shrimps and crabs, with a total of 32,735 samples, including 11,164 samples in 2019, 11,439 samples in 2020 and 10,132 samples in 2021. The national food safety standard maximum residue limits for veterinary drugs in foods (hereinafter referred to as the standards) specify the limit indicators for veterinary drug residues in freshwater products, of which the limit for florfenicol is 1000 μg/kg, the limit for enrofloxacin is 100 μg/kg and the limit for of sulfonamides is 100 μg/kg. In addition, the standards also specify the allowable daily intake of veterinary drugs in freshwater products, including florfenicol for 3 μg/(kg·d), enrofloxacin for 6.2 μg/(kg·d) and sulfonamides for 50 μg/(kg·d).

Data on the consumption of freshwater products by the population were obtained from the Fifth China Total Diet Study [17], which conducted a dietary questionnaire survey on the main food items consumed by residents of 20 Chinese provinces and estimated the consumption data using stratified multistage population-proportional whole-group random sampling.

#### 2.1.2. Data Preprocessing

Referring to the principles of credible assessment of low-level contaminants in food proposed by GEMS/FOOD, when the number of non-detected samples was less than 60% of the overall sample size, all non-detected data were replaced by 1/2 of the limit of detection (LOD); when the number of non-detected samples was higher than 60% of the overall number of samples, all non-detected data were replaced by the LOD [18]. Since the sample data of non-detected veterinary drug residues in this study were much less than 60%, all non-detected data were assigned the 1/2 LOD value for statistical calculation in this paper.

### 2.2. Safety Risk Assessment Model for Freshwater Products

Considering that the object of evaluation in this study was a single food and multiple contaminants, according to the risk assessment method and the set model use, based on the main influencing factors of health risk caused by food contaminants, this paper selected the Nemerow Integrated Pollution Index (NIPI), Index of Food Safety (IFS) and Hazard Risk Factor (R) as the three evaluation indicators of the freshwater product safety risk assessment model.

#### 2.2.1. Nemerow Integrated Pollution Index

The NIPI can reflect the characteristics of food contamination, taking into account the mean and maximum values of the single-factor contamination index, and it can highlight the role of the more contaminated pollutants, and it is often used to assess air [19,20], water environmental quality [21,22,23], heavy metal contamination in soil [24,25] and vegetables [26,27,28]. In this paper, the NIPI is used to calculate the integrated contamination index of veterinary drug residues in the sampled samples based on the sampling data of freshwater products in each province. The expression of the single factor contamination index is as follows:(1)$$P_{i,j}=\frac{C_{i,j}}{S_{j}}$$
where $P_{i,j}$ is the single factor pollution index of j types of veterinary drug residues in freshwater products in province i; $C_{i,j}$ is the detection value of j types of veterinary drug residues in freshwater products in province i (μg/kg); $S_{j}$ is the national limit standard for residues of j types of veterinary drug in freshwater products (μg/kg).
(2)$$PI_{i}=\sqrt{\frac{P_{max(i)}^{2}+P_{ave\left(i\right)}^{2}}{2}}$$
where $PI_{i}$ is the comprehensive pollution index of freshwater products in province i; $P_{max(i)}$ denotes the maximum value of the single pollution index in province i; $P_{ave\left(i\right)}$ denotes the average value of the single pollution index in province i.

#### 2.2.2. Index of Food Safety

The IFS is a method constructed by the International Codex Alimentarius Commission (CAC) and the World Health Organization (WHO) as an evaluation of the risk of exposure to food safety hazards and is often used to assess the risk of pesticide residues [29,30,31,32] and veterinary drug residues in meat, vegetables, fruits and edible mushrooms [33,34,35,36]. In this study, the IFS was used to assess the risk of exposure to hazards for veterinary drug residues in freshwater products. In addition, this paper used the point assessment model in the FAO/WHO Principles and Methods for Risk Assessment of Chemicals in Foods for dietary exposure assessment, with the following expressions:(3)$$EDI_{i,j}^{50}=\frac{F_{i}^{50}\times C_{avg\left(i,j\right)}}{W}$$
where $EDI_{i,j}^{50}$ is the average daily intake of j types of veterinary drug residues per kilogram of body weight of the population in province i through freshwater products (μg/kg·bw·d); $F_{i}^{50}$ is the average consumption of freshwater products in province i (kg/d); $C_{avg\left(i,j\right)}$ is the average detection value of j types of veterinary drugs residues in freshwater products in province i (μg/kg); W is the target intake population average weight (kg), taken as 60 kg.
(4)$$IFS_{i}=\underset{j}{\sum }\frac{EDI_{i,j}^{50}\times f}{SI_{j}}$$
where $IFS_{i}$ is the food safety index of province i; f is the correction factor for safe intake, and the value of 1 is taken from the relevant literature [37]; $SI_{j}$ is the safe daily intake (μg/kg·bw·d) of j types of veterinary drug residues in freshwater products, using ADI.

#### 2.2.3. Hazard Risk Factor

The hazard risk factor takes into account the influence of the exceedance rate or positive detection rate of the hazard, the frequency of administration and its own sensitivity, and it provides an intuitive and comprehensive reflection of the risk level of the hazard over time, and it is used by researchers to assess the risk system of pesticide and veterinary drug residues in vegetables and other foods [38]. In this paper, the hazard risk factor is used to assess the hazard risk coefficient of veterinary drug residues in freshwater products with the following expression:(5)$$R_{i}=\underset{j}{\sum }(aH_{i,j}+\frac{b}{F_{i,j}}+S_{j})$$
where $R_{i}$ is the hazard risk factor of freshwater products in province i, and $H_{i,j}$ is the exceedance rate of j types of veterinary drug residues in freshwater products in province i. $F_{i,j}$ is the frequency of the administration of j types of veterinary drug residues in freshwater products in province i. This target contaminant was a mandatory item; therefore, $F_{i,j}$ was taken as 1; $S_{j}$ is the sensitivity factor of the contaminant, which can be adjusted appropriately according to the sensitivity and importance of the current concern of the hazard in the food safety domain. In this study, the target contaminants were all normally administered hazards, and $S_{j}$ was taken as 1; a and b are the corresponding weight coefficients, respectively, and in order to make the risk coefficient $R_{i}$ reflect the effects of $H_{i,j}$ and $F_{i,j}$ in an accurate and balanced way, a is usually taken as 100 in practical applications, while b is taken as 0.1 [37].

### 2.3. Freshwater Product Safety Risk Classification Based on K-Means

To reduce the influence of a single factor in food safety risk assessment, this paper integrates NIPI, IFS and R indicators to construct a risk assessment model of veterinary drug residues in freshwater products. To eliminate the influence of subjective factors in risk grading, this paper uses a clustering algorithm to grade the safety risk of freshwater products in different provinces at different time periods, and it constructs the safety risk grading space based on the risk assessment model. Since the amount of data in this subject is small and there is no dirty data, the k-means [39] algorithm is fast and efficient and can achieve a good clustering performance on sample spaces of arbitrary shapes, which was suitable for analyzing the model data of this study, so the k-means algorithm was selected for food safety risk grading in this paper. The specific process of the algorithm is as follows:(1)First, any k values are selected in the data set as the initial center of mass.(2)Calculate the distances of each of the remaining points to these k centers of mass in turn, and divide each point into clusters with the center of mass that is closest to it.(3)Obtain k clusters in this way, and then calculate the mean value of these k clusters as the new center of mass.(4)Repeat the steps (2) and (3) until the cluster centers no longer change or the number of iterations is reached; then, the safety risk grading space construction is completed and the algorithm converges.

### 2.4. Transformer-Based Model for Predicting the Safety Risk Level of Freshwater Products

#### 2.4.1. Freshwater Product Safety Risk-Level Prediction Process

In this study, after data cleaning, integration, transformation and normalization, there were 3180 sample points in the sample space, which is a small sample dataset. Deep learning algorithms, such as Recurrent Neural Network (RNN), Long Short-Term Memory (LSTM) and Gated Recurrent Units (GRU) algorithms, are widely used in various industries for prediction and analysis, but a large number of datasets are required for training when building models. Therefore, in order to improve the accuracy of safety risk-level prediction, this paper constructed a Transformer-based freshwater product safety risk-level prediction model. The Transformer neural network model was improved to suit the application scenario in this paper, and the details are described in Section 2.4.2.

The Transformer-based freshwater product safety risk-level prediction model proposed in this paper is shown in Figure 1, and the model is divided into three layers, which are the data layer, Transformer prediction layer and risk-level prediction layer.

First, at the data layer, the sampling data and consumption data of veterinary drug residues of freshwater products in each province are used to construct freshwater product safety risk assessment indicators, including NIPI, IFS and R, and based on the above safety risk assessment indicators, the weekly freshwater products in each province are classified into a risk level by the k-means algorithm to construct the risk-level space and complete the experimental data set construction. We put each risk indicator at the moment of T into the Transformer prediction layer to wait for predicting each risk indicator at the moment of T + 1.

Second, in the Transformer prediction layer, this paper uses the Transformer [40,41,42,43] algorithm to predict each safety risk assessment indicator of veterinary drug residues in freshwater products in each province, using the multi-layer encoder–decoder mechanism in the Transformer, combined with multi-headed attention to improve the accuracy of prediction.

Finally, in the risk-level prediction layer, the predicted value of each safety risk assessment indicator at the moment of T + 1 is output from the full linkage layer, and the distance between this safety risk assessment indicator and the clustering center that has been divided is measured, and the freshwater product risk level of the province for that week is categorized into the cluster with the closest distance.

#### 2.4.2. Transformer-Based Predictive Model for Freshwater Product Safety Risk Assessment Indicators

In this study, the food safety risk assessment indicators are the contamination indicators of freshwater products in each province over a period of time, which are time-series sequences, so the model needs to have the ability to model long-term memory. Therefore, this paper improves the Transformer structure according to the food safety risk assessment indicator prediction application scenario, as shown in Figure 2.

The core of the Transformer-based prediction model for food safety risk assessment metrics lies in its encoder and decoder structures, both of which consist of six identical layers stacked on top of each other, where each layer of the encoder contains two sub-layers of a multi-headed self-attention mechanism and feedforward neural network, and each layer of the decoder contains three sub-layers of a masked multi-headed self-attention mechanism, encoder–decoder multi-headed attention mechanism and feedforward neural network.

First, we construct the input food safety risk assessment indicator matrix, letting the time window be t and the current moment be T, and need to predict the safety risk assessment indicator at the moment of T + 1; then, we put the indicator matrix $X=\left[X_{T-t},\cdots ,X_{T-1},X_{T}\right]$ into the encoder. The Transformer benefits from a number of advantages thanks to its purely attentional mechanism construct, but this deprives it of the ability to learn sequence position information. In the food safety risk assessment indicators prediction scenario, the position information of the vectors in the matrix represents the moment information, which plays a crucial role in the assessment indicator prediction. To address this issue, a position-encoding operation was added to the input matrix of the encoder and decoder to integrate the position information into the input sequence, as shown in Equations (6) and (7).
(6)$$P_{\left(p,2i\right)}=sin\left(\frac{p}{10,000^{\frac{2i}{d_{model}}}}\right)$$
(7)$$P_{\left(p,2i+1\right)}=\cos\left(\frac{p}{10,000^{\frac{2i}{d_{model}}}}\right)$$
where p denotes the position of the indicator vector, $d_{model}$ denotes the dimension of the indicator vector and the position of each indicator vector is encoded by the cosine and sine function of different frequencies.

The encoder is responsible for encoding the input evaluation indicator matrix and mapping it into an intermediate vector containing the input information, the core principle of which is the self-attention mechanism. The self-attention mechanism is a variation of the attention mechanism, which reduces the reliance on external information and is better at capturing the internal relevance of the data. Its purpose is to filter out a small amount of important information from the input evaluation indicator matrix and use weights to represent the importance of the information so that the model focuses on the more important information. The self-attentive mechanism uses scaled dot product attention to calculate the attention value of the indicator matrix by first performing dot product and SoftMax normalization on the query matrix and key matrix to calculate the weight coefficients and then weighting and summing the value matrix according to the weight coefficients, as shown in Equations (8)–(11).
(8)$$Attention\left(Q,K,V\right)=softmax\left(\frac{QK^{T}}{\sqrt{d}}\right)V$$
(9)$$\;Q=XW^{Q}$$
(10)$$K=XW^{K}$$
(11)$$V=XW^{V}$$
where Q is the query matrix, K is the key matrix and V is the value matrix. These three matrices are obtained by multiplying the input indicators matrix X with the corresponding weight matrices $W^{Q}$, $W^{K}$ and $W^{V}$, respectively, and d is the dimensionality of Q, K and V.

In order to synthesize the information contained in the input matrix, the self-attentive mechanism uses multi-headed self-attentive mechanisms to jointly focus on the information from different manifestation subspaces at different locations. The multi-headed self-attentive mechanism splices multiple self-attentive mechanisms and uses multiple self-attentive heads to learn the information from different performance subspaces separately, and then, it splices and linearly transforms multiple attention values to obtain the final attention values to achieve the modeling expression of different constraints, as shown in Equations (12) and (13).
(12)$$h_{i}=Attention\left(XW_{i}^{Q},XW_{i}^{K},XW_{i}^{V}\right)$$
(13)$$MultiHead\left(Q,K,V\right)=Concat\left(h_{1},\ldots ,h_{m}\right)\cdot W$$
where $W_{i}^{Q}$, $W_{i}^{K}$ and $W_{i}^{V}$ are the weight matrices of the *i*th attention head Q, K and V. W is the multi-head attention weight matrix, m is the number of attention heads, and the Concat function is used to splice the output values calculated by each attention head.

The decoder is responsible for decoding the intermediate vector output from the encoder into the output sequence, and its core principles are the encoding–decoding multi-headed attention mechanism and the masking multi-headed self-attentiveness mechanism. In order to improve the accuracy of evaluation indicator prediction, in addition to learning the dependency between input feature sequences in the multi-headed self-attention mechanism, the dependency between input feature sequences and shift sequences should also be considered, so the encoding–decoding multi-headed attention mechanism is used in the decoder. The input of the blocking multi-headed self-attention mechanism module is the shift sequence, and its purpose is to use the multi-headed self-attention mechanism to learn the dependencies between the shift sequences and input the dependencies into the encoding-decoding multi-headed attention mechanism module, so that the whole Transformer-based food safety risk assessment index prediction model can comprehensively learn the dependencies between the input feature vectors and their mutual dependencies.

In order to solve the problem that increasing the depth of the network instead affects the prediction accuracy of food safety risk assessment indicators, a residual connection operation was added between each sub-layer of the encoder and decoder to focus on the change of the difference part before and after training and to improve the training effect. At the same time, in order to accelerate the network convergence and improve the network generalization ability, each sub-layer also adopts the layer normalization operation at the same time, as shown in Equation (14).
(14)o=LayerNorm(x+Sublayer(x))
where Sublayer is the individual attention mechanism layer processing function and the fully connected feedforward neural network processing function. LayerNorm is the layer normalization processing function.

Finally, a layer of the fully connected network is added to the output to output the predicted indicators of the safety risk level at the next moment and participate in the construction of the input indicator matrix for the next time.

## 3. Results

### 3.1. Data Set and Experimental Environment

#### 3.1.1. Data Set

In this paper, three food safety risk assessment indicators for freshwater products in each province will be predicted separately, and the total length of the time series of freshwater products in each province was 159 weeks in the experiment. The pre-processed data set will be divided into the training set and test set, and the ratio will be 6:4 according to the number of data entries.

#### 3.1.2. Experimental Environment

In this paper, the open source PyTorch [44] deep learning framework was used as the experimental platform, and the specific parameters of the experimental environment are shown in Table 1.

### 3.2. Model Evaluation Indexes

The safety risk level of freshwater products is determined by the above three indicators together, so this paper evaluated the single performance of each of the three indicators and tested the accuracy of the predicted safety risk level.

#### 3.2.1. Performance Evaluation Indexes for Single Indicator Prediction

In this paper, the Mean Absolute Percentage Error (MAPE) and Mean Squared Error (MSE) are used to evaluate the forecasting efficacy of NIDI, IFS and R in the proposed model. These two indexes are calculated as follows:(15)$$\mathrm{MAPE}=\frac{1}{n}\overset{n}{\underset{i=1}{\sum }}\left|\left(y_{i}-\hat{y_{i}}\right)/y_{i}\right|$$
(16)$$\mathrm{MSE}=\frac{1}{n}\overset{n}{\underset{i=1}{\sum }}{\left(y_{i}-\hat{y_{i}}\right)}^{2}$$
where $y_{i}$ is the actual value of a single assessment indicator in week i, and $\hat{y_{i}}$ is the predicted value of a single assessment indicator in week i.

#### 3.2.2. Accuracy Assessment Indexes for Risk-Level Prediction

In this paper, three assessment indexes, precision rate, recall rate and F-measure value, are used to test the accuracy of risk-level prediction.
(17)$$P=\frac{TP}{TP+FP}$$

In the precision rate Equation (17), TP represents the number of samples for which the model correctly predicts the risk level, and FP represents the number of samples for which the model predicts not that risk level as that risk level.
(18)$$R=\frac{TP}{TP+FN}$$

In the recall Equation (18), TP represents the number of samples for which the model correctly predicts the risk level, and FN represents the number of samples for which the model predicts that risk level as other risk levels.
(19)$$F-Measure=\frac{2*P*R}{P+R}$$

To better evaluate the performance of the prediction model, this paper uses the F-Measure score as an evaluation criterion to measure the comprehensive performance of the model, as shown in Equation (19).

### 3.3. Freshwater Product Safety Risk Assessment and Classification

#### 3.3.1. Freshwater Product Safety Risk Assessment Indicators

In order to comprehensively assess the hazards of veterinary drug residues in freshwater products, the weekly NIPI, IFS and R values in freshwater products in each province from 2019 to 2021 were calculated and obtained based on the hierarchical analysis, and the set of three food safety risk assessment indicators for three years in 20 provinces is shown in Figure 3, Figure 4 and Figure 5.

#### 3.3.2. Risk Classification

According to Figure 3, Figure 4 and Figure 5, it can be seen that different safety risk assessment indicators of freshwater product refer to a large difference in the order of magnitude, in order to avoid the disparity in the number of indicators affecting the assessment effect; therefore, this paper used Z-Score data standardization for the three indicators, as shown in Equation (20). Where $Index_{scale}$ denotes the standardized indicator, $Index_{org}$ denotes the original indicator, $mean_{org}$ denotes the mean of the indicator and $std_{org}$ denotes the standard deviation of the indicator.
(20)$$Index_{scale}=\frac{Index_{org}-mean_{org}}{std_{org}}$$

In this paper, the k-means algorithm is used for clustering, and NIPI, IFS and R are selected as clustering features. Figure 6 shows the line graph of the silhouette coefficient of the number of clusters from 2–7. It can be seen from the figure that the silhouette coefficient is the largest when the clustering result is three clusters and is much larger than the other clusters, indicating that the instances within the clusters are more compact and that the inter-cluster distance is larger when the clusters are three clusters. Therefore, this paper divides the risk level into three levels, and the normalized clustering center, risk classification and the number of samples in each level are shown in Table 2. The distance of the cluster center from the origin is calculated according to the normalized index, and the risk levels of categories 1–3 are defined as low–medium–high, respectively. From Table 2, it can be seen that the indicators of clustering centers increase sequentially with the increase of risk grades.

#### 3.3.3. Analysis of Risk Grading Results

It can be analyzed from Figure 7, Figure 8 and Figure 9 that the distribution of actual indicators for different safety risk levels of freshwater products is as follows:(1)Characteristics of category 1: NIPI intervals are relatively small, with intervals concentrated in 0; IFS concentrates in 0; R concentrates in 3.3.(2)Characteristics of category 2: NIPI interval is relatively large, with interval distribution in 0–0.5; IFS concentrates in 0–1; R is distributed in 3.8–4.5.(3)Characteristics of category 3: NIPI interval is the largest, with interval distribution in 1–4; IFS concentrates in 5–10; R is distributed in 3–5.(4)Comparative analysis: the NIPI interval of category 1 is short, and the values are generally small; the IFS values are distributed in a range with a small average value and a small distribution, and the R values are small and stable, but the distribution density values of the indicators are large, which is a class of freshwater products with a small-risk level. The NIPI, IFS and R values of category 2 are at a medium level, which is a type of freshwater product at a medium-risk level. Category 3 has a large interval and relatively large value of NIPI, IFS is at a high level and the average value of R value is also large, and the distribution density of indicators is small, which is a freshwater product at a high-risk level and deserves key attention.

In the clustering results, we have found that the provinces with higher risk levels are mainly Guangdong, Guangxi, Hebei, Henan, Hubei, Hunan, Jiangxi, Shanghai, Sichuan and Zhejiang, and the seasons with higher risk levels are mainly concentrated in spring and summer. Because spring is a season of continuous warming, after a winter of frozen water, fish and other biological metabolites accumulate in the bottom of the pond water; at the same time, a large number of pollutants in the air with the snow are deposited on the ice of the fish pond and then into the water, resulting in a variety of toxic and harmful substances that may exist in the water, when the fish are poor, and it may be accompanied by a variety of diseases; the summer temperature is higher, but the water temperature is relatively suitable, and it is the peak season of fish growth, and the water is easy to breed and multiply various pathogens; it is the fish susceptible to disease season, but it is also the most difficult period of medication and management. At the same time, the data show that veterinary drug residues that exceed the standard occur mostly in the provinces with richer water resources but also in the large breeding provinces. Accordingly, it is necessary to impose supervision on key provinces and key months.

### 3.4. Prediction Results of Freshwater Product Safety Risk Level Based on a Transformer

In order to demonstrate the effectiveness of a Transformer-based model for predicting the safety risk level of freshwater products, RNN, LSTM and GRU prediction models, which are commonly used today, were selected and compared with the model proposed in this paper. Firstly, the weekly NIPI, IFS and R indicators of freshwater products in each province were predicted separately. The risk level of the province for that week was classified according to the predicted food safety risk assessment indicators. Figure 10, Figure 11, Figure 12, Figure 13 and Figure 14 show the three assessment indicators for each of the 20 provinces predicted by the model proposed in this paper, with a time step of 7. Since the effectiveness of risk-level prediction is directly determined by the results of indicator prediction, the following statistical analysis of the prediction results of three risk assessment indicators was conducted using MAPE and MSE evaluation indicators.

In Figure 10, Figure 11, Figure 12, Figure 13 and Figure 14, the pink line indicates the actual set of values for the indicators, the blue line indicates the set of values that are used to train the model and the purple line indicates the results predicted by the indicators. Weeks 0–137 were the training set of the model, and weeks 138–159 were the testing set of the model. From the experimental results, it can be seen that most of the predicted curves matched with the actual curves, and very few values did not match with the actual predictions. It was found that there are two main reasons for this discrepancy between predictions and actuals; on the one hand, it is due to the change of the supply chain caused by unexpected events, and on the other hand, it is due to the strengthening of government regulation, which can reduce the pollution index of freshwater products.

Figure 15 and Figure 16 show the MAPE and MSE values of the three food safety risk assessment indicators of freshwater products predicted by the four models, respectively. From the experimental result plots, it can be seen that the prediction models proposed in this paper predicted the three indicators with the smallest MAPE and MSE values, which performed better than the other models. It is generally believed that the prediction accuracy is higher when MAPE is less than 10. The MAPE of the three indicators predicted by the Transformer proposed in this paper was less than 10, and the prediction effect was good, among which the MAPE of the R indicator was the smallest, at only 0.0156, while the MAPE of the NIPI indicator was the largest. Meanwhile, it could be seen that among the four models, the effect of RNN prediction of each indicator deviated more from the correct value and fluctuated more. LSTM and GRU also had a better prediction effect on IFS and R indicators, but the prediction deviated more from the correct value on the NIPI indicator.

After predicting the NIPI, IF, and R values for a single week in each province by the above four neural network models, the distance between these integrated assessment indicators and the three clustering centers was measured, and the risk level of that province for that week was classified into the nearest cluster to determine that risk level. The precision, recall and F-measure of the risk level predicted by the four models were tallied, as shown in Table 3.

The experimental results show that the Transformer-based prediction model was significantly better than the other three models in terms of recall rate. Since the recall rate was as complete as possible to predict all possible risks, this model can provide a good data basis for the government’s comprehensive attention and regulation. At the same time, the model also performed well in terms of precision, which can provide clues for the government to grasp the areas and foods that may have hidden risks. In addition, the F-Measure statistic shows that the model proposed in this paper had a good balance between recall and precision, which provides an effective tool for the government to focus on hierarchical regulation.

## 4. Discussion

Excessive veterinary drug residues have a major impact on food safety, not only endangering human health, but also affecting social development. China’s vast land resources and abundant products make it a waste of resources if all products are regulated universally throughout the year, and when food safety incidents break out and are then dealt with, they can cause serious public opinions and consequences. Therefore, it is important to focus on the supervision of freshwater products containing veterinary drug residues and early warning. This paper constructs a Transformer-based model for predicting the risk level of veterinary drug residues in freshwater products in China, which provides a systematic risk measurement method for the government and a scientific reference basis for confirming the priority of regulation in supervision, as well as technical support for government intervention. The experimental results show that the prediction model proposed in this paper predicts a recall rate of 94.14%, which can meet the needs of food regulators and reassure consumers while allowing producers to benefit from ensuring food safety and improving quality. The proposed approaches in the paper can combine other parameter estimation algorithms [45,46,47] to study the temporal prediction problems of nonlinear systems with different disturbances [48,49,50,51], and can be applied to other fields such as visual processing and engineering application systems [52,53,54]. 

## Figures and Tables

**Figure 1 foods-11-01690-f001:**
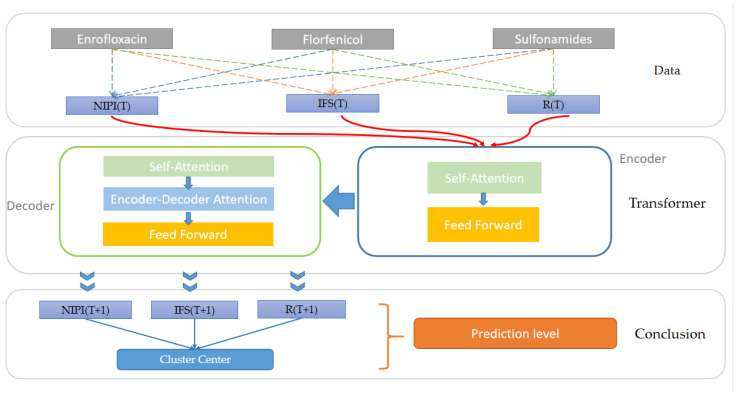
Transformer-based model for predicting the safety risk level of freshwater products.

**Figure 2 foods-11-01690-f002:**
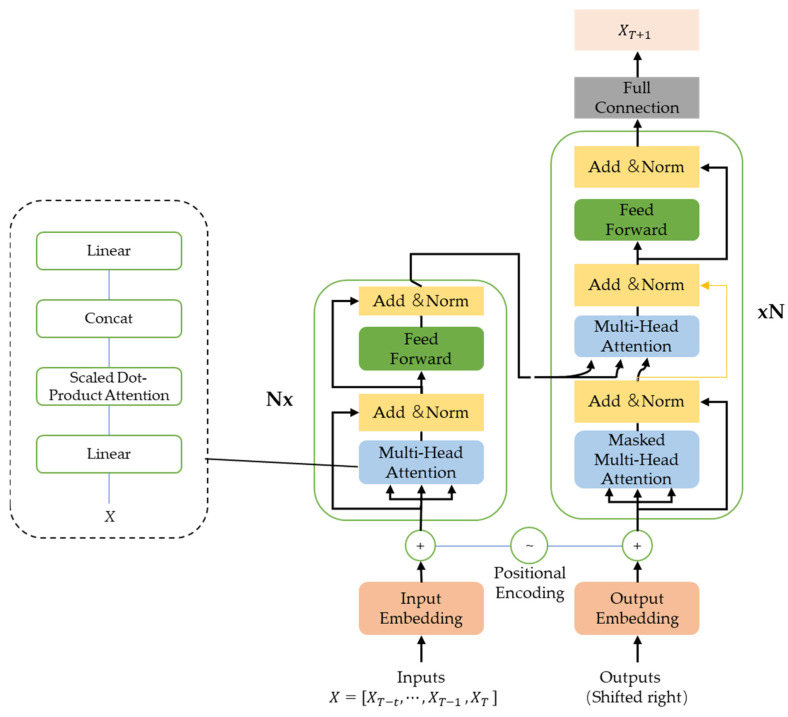
Transformer-based predictive model for freshwater product safety risk assessment indicators.

**Figure 3 foods-11-01690-f003:**
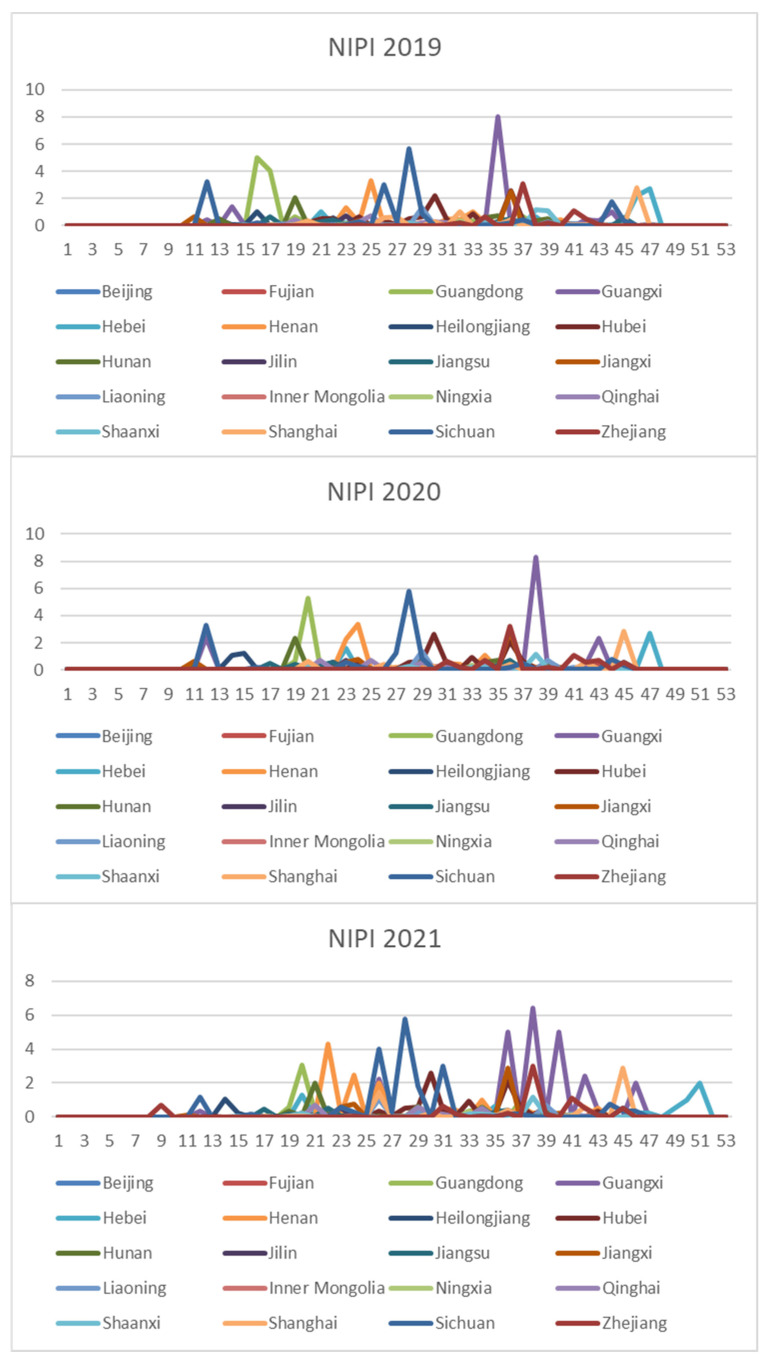
Collection of weekly NIPI indicators for freshwater products in 20 provinces from 2019 to 2021.

**Figure 4 foods-11-01690-f004:**
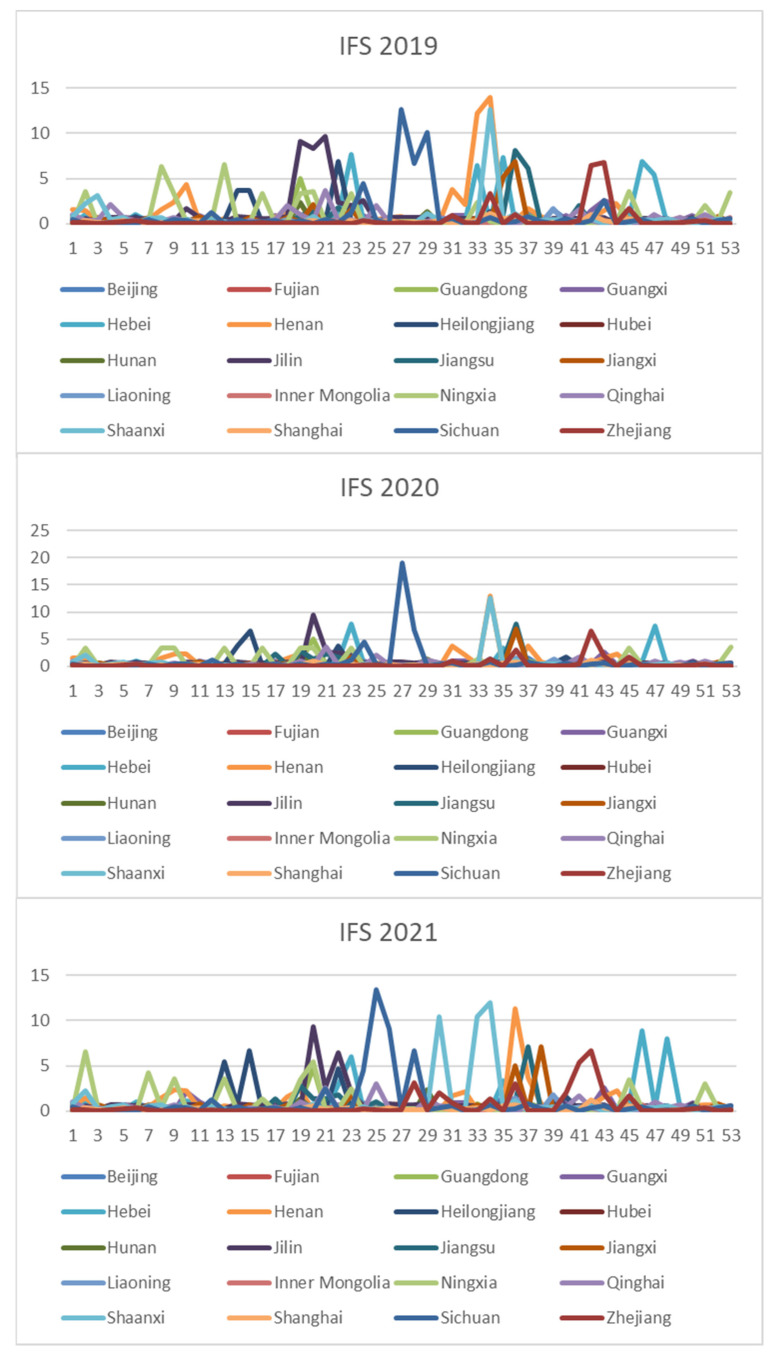
Collection of weekly IFS indicators for freshwater products in 20 provinces from 2019 to 2021.

**Figure 5 foods-11-01690-f005:**
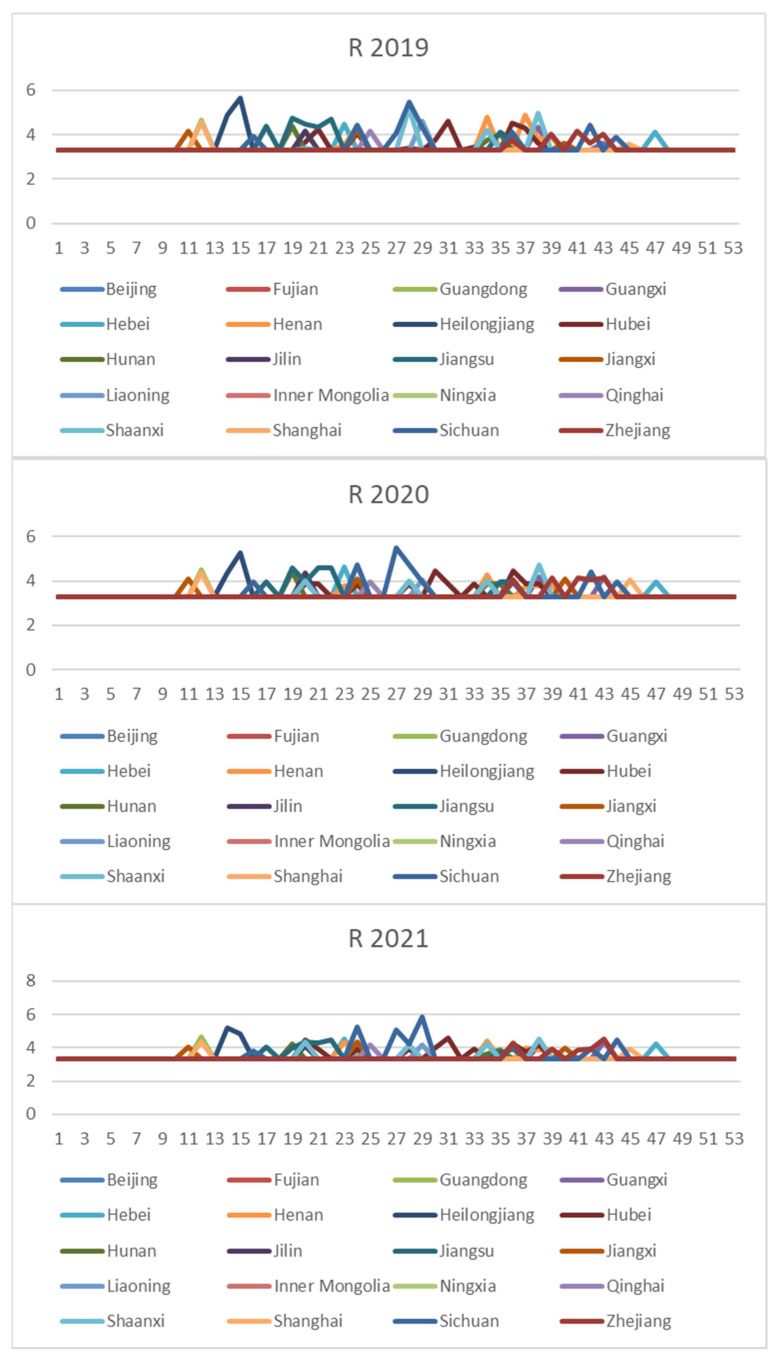
Collection of weekly R indicators for freshwater products in 20 provinces from 2019 to 2021.

**Figure 6 foods-11-01690-f006:**
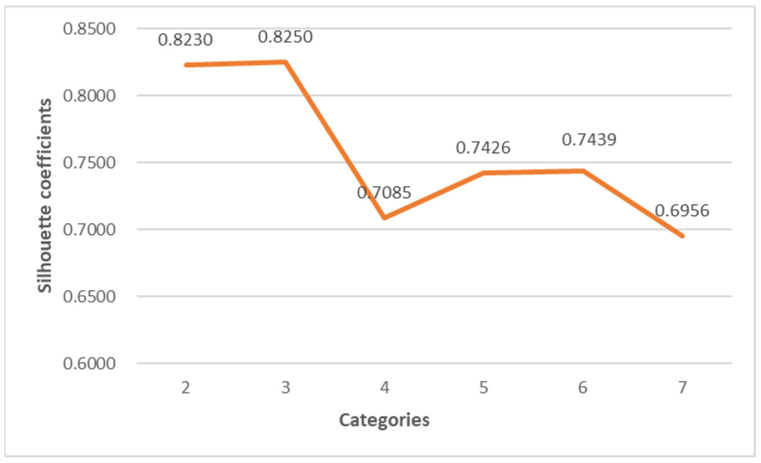
Silhouette coefficients of five types of clustering category.

**Figure 7 foods-11-01690-f007:**
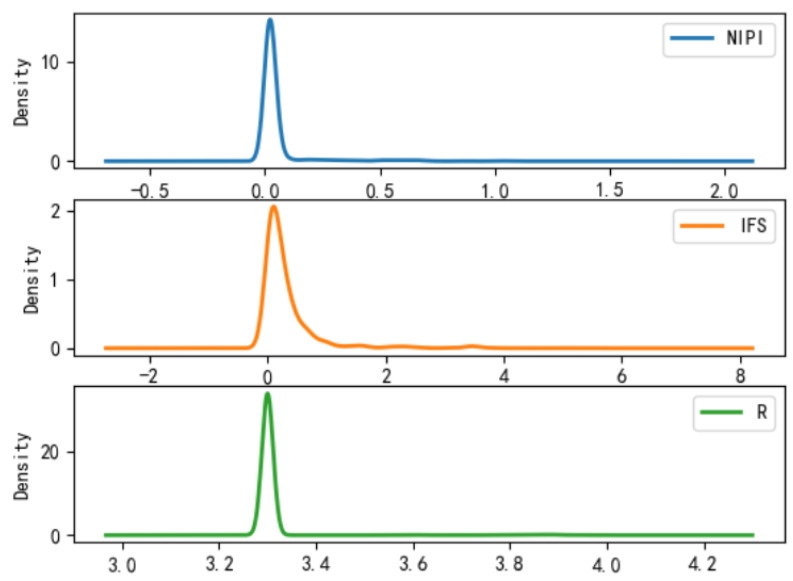
Probability density function plot for category 1.

**Figure 8 foods-11-01690-f008:**
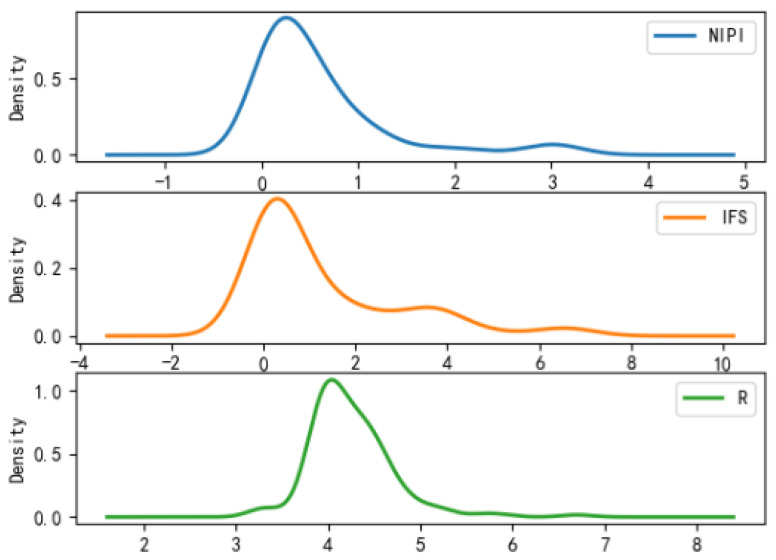
Probability density function plot for category 2.

**Figure 9 foods-11-01690-f009:**
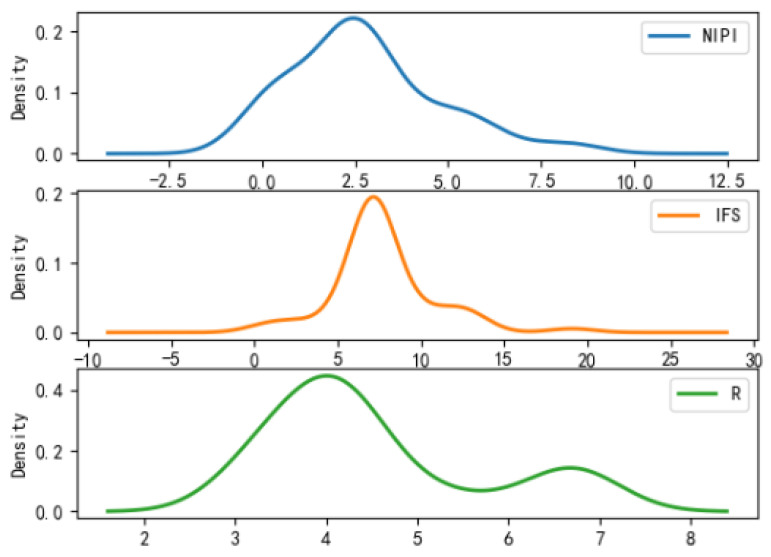
Probability density function plot for category 3.

**Figure 10 foods-11-01690-f010:**
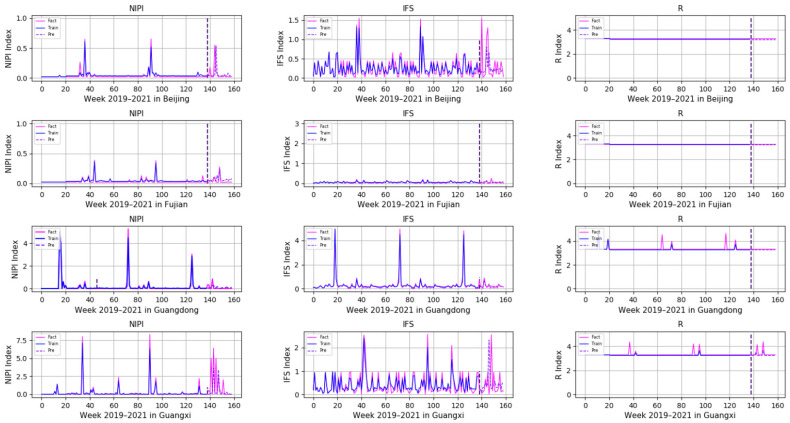
Predicted results of NIPI, IFS and R indicators in Beijing, Fujian, Guangdong and Guangxi.

**Figure 11 foods-11-01690-f011:**
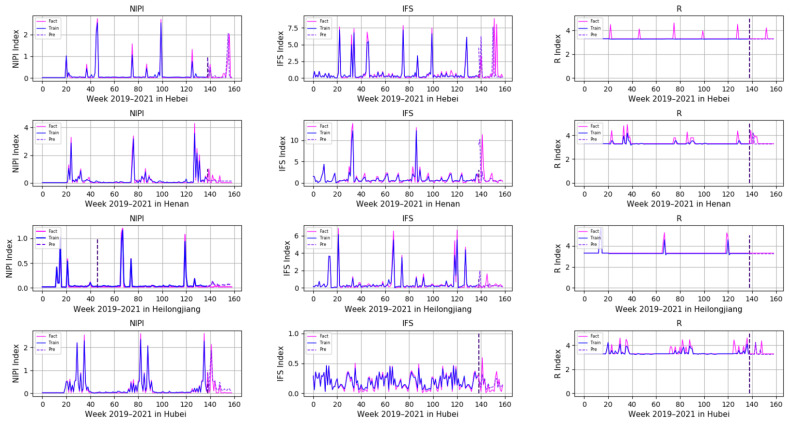
Predicted results of NIPI, IFS and R indicators in Hebei, Henan, Heilongjiang and Hubei.

**Figure 12 foods-11-01690-f012:**
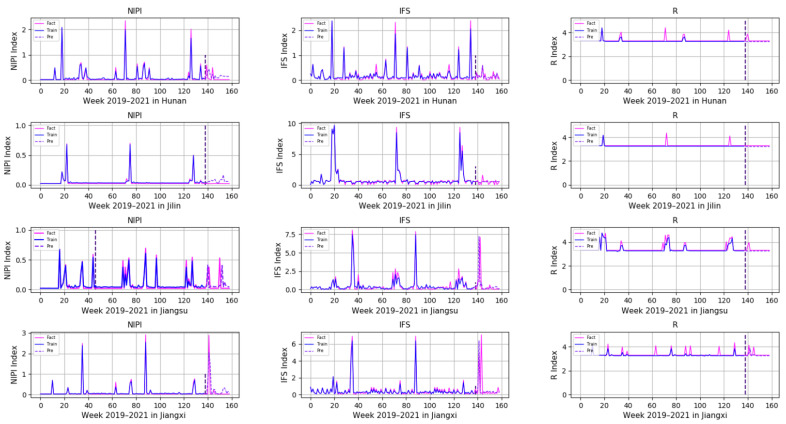
Predicted results of NIPI, IFS and R indicators in Hunan, Jilin, Jiangsu and Jiangxi.

**Figure 13 foods-11-01690-f013:**
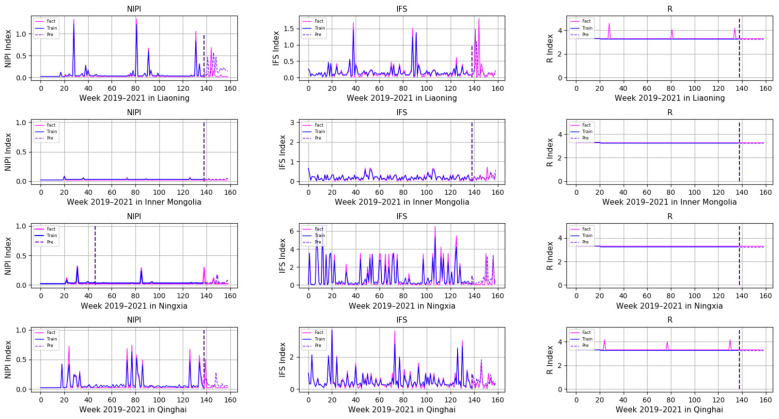
Predicted results of NIPI, IFS and R indicators in Liaoning, Inner Mongolia, Ningxia and Qinghai.

**Figure 14 foods-11-01690-f014:**
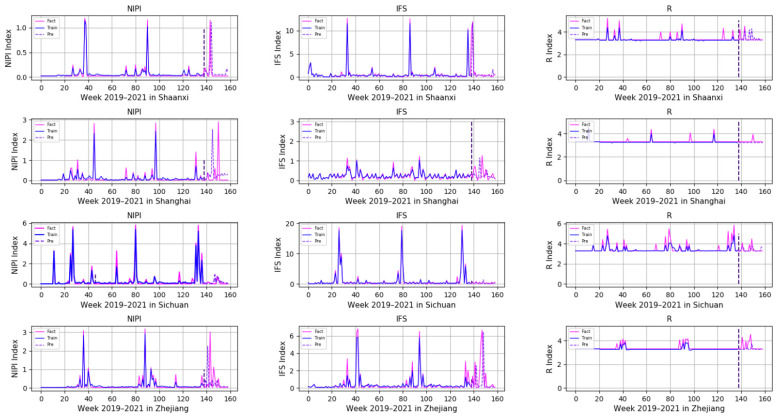
Predicted results of NIPI, IFS and R indicators in Shaanxi, Shanghai, Sichuan and Zhejiang.

**Figure 15 foods-11-01690-f015:**
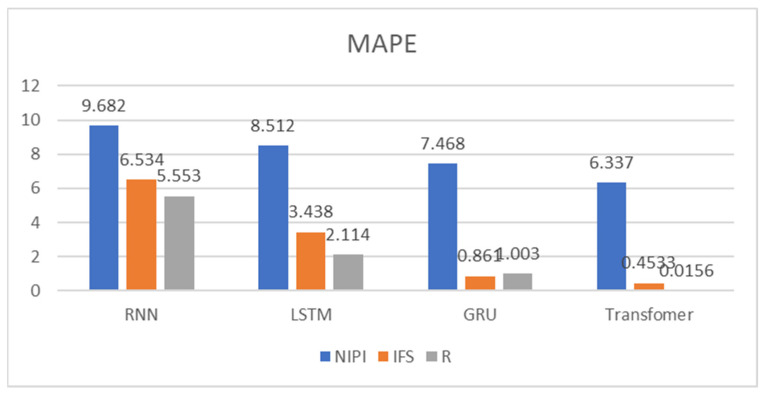
MAPE for NIPI, IFS and R indicators.

**Figure 16 foods-11-01690-f016:**
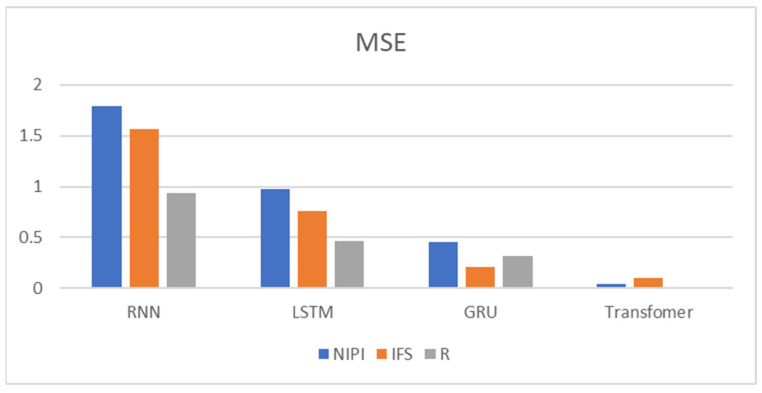
MSE for NIPI, IFS and R indicators.

**Table 1 foods-11-01690-t001:** Experimental environment parameters.

**Operating System**	Windows 10	64-bit
**Hardware information**	CPU	Intel CORE i7-9700F@3.00GHz eight-core
	GPU	Nvidia GeForce RTX3060
	RAM	16 GB
**Software tool**	Python 3.7.11	Numpy 1.18.5
		Pandas 1.2.2
		Torch 1.11.0
		Matplotlib 3.3.3

**Table 2 foods-11-01690-t002:** Clustering centers and ranking of the three clusters.

Category	NIPI	IFS	R	Sample Size	Risk Level
1	−0.139858	−0.129344	−0.193611	2997	Low
2	0.990525	0.612015	2.905360	133	Medium
3	5.459207	5.677968	3.782895	50	High

**Table 3 foods-11-01690-t003:** Statistical results of the accuracy of risk-level prediction.

Model	Index-Data		
	P%	R%	F-Measure%
RNN	80.16	80.29	80.22
LSTM	82.46	82.77	82.61
GRU	87.94	88.31	88.12
Transformer	93.73	94.14	93.93

## Data Availability

Restrictions apply to the availability of these data. Data were obtained from the State Administration for Market Regulation Statistics and are available at [55] with the permission of the State Administration for Market Regulation Statistics.

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
