# Peer review of "Prediction of Safety Risk Levels of Veterinary Drug Residues in Freshwater Products in China Based on Transformer"

_foods, 2022, doi:10.3390/foods11121690_

Round 1

Reviewer 1 Report

The importance of the subject is clear, since the concerns about anthropogenic contaminants are increasing, even among consumers. The study is very complete. However, some comments can be made in order to clarify the work presented.

First of all, please simplify/clarify the title.

It is stated that “his paper investigates the national sampling data of veterinary drug residues in freshwater products in 2020, and selects the three veterinary drug residues with the highest sampling levels in freshwater products for assessment and prediction”. The authors believe that the analysis of only one year can provide the real and complete evidence of the medicines used?

Considering the antibiotics analyzed (ofloxacin, enrofloxacin and sulfonamides) is it clear that those are the main used in China? Any evidence on that?

Lines 107-109: Provide references for the limits presented.

Reviewer 2 Report

In this manuscript, the authors aim to investigate the national sampling data of veterinary drug residues in freshwater products in 2020 with three veterinary drug residues with the highest sampling levels in freshwater products for assessment and prediction, including ofloxacin, enrofloxacin and sulfonamides.

The subject of the manuscript is original and falls within the scope of the journal. The authors present a well-written manuscript describing a logical and well-organized study.

1.      Although many different veterinary drugs belonging to different groups are used in fishery products, it is not clearly understood why these 3 antibiotics were chosen. Moreover, this does not fully reflect the title of the article. “Veterinary Drug Residues” or “Antibiotic residues”?

2.      While risk assessment or hazard risk should be done separately according to the type of veterinary drugs and fisheries used, the scientific basis for the combined evaluation of different drugs and products has been evaluated as insufficient.

3.      It should be stated in the article which seafood product contains what level of residue and which analytical method these residues are determined by.

4.      Reference list:  Please check the consistency of the reference list for the guidelines of the journal (The journal names of the references should be in the same format, ref 15 should be revised (title?)).

5.  Lines 505-511: The conclusion sentence is too long. This statement should be revised to be short and clear.
